# Physical Properties of Chitosan Films Containing Pomegranate Peel Extracts Obtained by Deep Eutectic Solvents

**DOI:** 10.3390/foods10061262

**Published:** 2021-06-02

**Authors:** Aikaterini Kyriakidou, Dimitris P. Makris, Athina Lazaridou, Costas G. Biliaderis, Ioannis Mourtzinos

**Affiliations:** 1Department of Food Science and Technology, Faculty of Agriculture, Aristotle University of Thessaloniki, 54124 Thessaloniki, Greece; kfnskyriakidou@gmail.com (A.K.); athlazar@agro.auth.gr (A.L.); biliader@agro.auth.gr (C.G.B.); 2Department of Food Science & Nutrition, School of Agricultural Sciences, University of Thessaly, N. Temponera Street, 43100 Karditsa, Greece; dimitrismakris@uth.gr

**Keywords:** deep eutectic solvents, pomegranate peel, extraction, polyphenols, chitosan films

## Abstract

Pomegranate peel is a byproduct of pomegranate juice production, and is rich in polyphenol compounds. The objective of this study was to investigate the incorporation of pomegranate peel extract in chitosan films. Green deep eutectic solvents (DESs) were used as extraction solvents. Choline chloride (ChCl) and glycerol (Gly) were used as the hydrogen bond acceptor and hydrogen bond donor, respectively; the molar ratio of the DES ingredients, ChCl:Gly, was 1:11. The extraction process was optimized by deploying response surface methodology. Under the optimized conditions, the extraction yield in total polyphenols amounted to 272.98 mg of gallic acid equivalents per g of dry matter and, for total flavonoids, 20.12 mg of quercetin equivalents per g of dry matter, with a liquid to solid ratio of 47 mL g^−1^, time of 70 min, and 30% (*v*/*v*) water concentration in the DES. Afterwards, composite chitosan films were prepared by using five different formulations; the DES containing extract was incorporated as a plasticizer in the chitosan films. Specimens of every recipe were submitted to large deformation tensile testing in Texture Analyzer. Furthermore, water sorption behavior and color parameters of the films were determined.

## 1. Introduction

Pomegranate is a fruit known for its high total phenolics content [1,2]. Recent studies have shown that extracts prepared from different parts of the pomegranate plant have various biological activities, including antimicrobial, antiviral, antioxidant, anti-inflammatory, anti-diabetic, cardioprotective, and anticarcinogenic. The therapeutic potential of pomegranate has been attributed to a variety of phytochemicals, such as tannins, ellagic acid, gallic acid, catechins, flavonoids, and anthocyanins. Several studies have shown that the pomegranate peel contains higher amounts of biologically active compounds than the edible pulp [3,4,5,6]. In the food industry, pomegranate is considered a multipurpose raw material, as it is used for the production of juices, marmalades, and syrups, as well as cosmetic creams and lotions, nutritional supplements, colorants, and food additives [7,8].

The pomegranate peel accounts for almost 30% of the fruit, and displays a remarkable antioxidant capacity because of its high content of polyphenolic compounds, such as punicalagin (an ellagitannin), gallic acid, and ellagic acid [9]. However, concerns have been raised regarding the proper utilization of the pomegranate peel by the industry, since 9 million tons of byproducts, corresponding to the production of ca. 1 million tons of concentrated pomegranate juice, are discarded without being further used [10]. Thus, research is now focused on finding alternative uses-valorization for this waste material in the industry to take advantage of its beneficial health bioactivities [11,12].

Common solvents often used over the last decade in the extraction of phenolics from pomegranate peels are water, methanol, ethanol, and their mixtures. The results obtained for the yield of total phenolics content in the pomegranate peel extracts varied depending on the solvent used, as well as the extraction method adopted. Such differences have been attributed to several factors, e.g., type and variety (cultivar) of the plant material, environmental conditions during cultivation, and the polarity of the solvents [4].

In the context of green chemistry, extraction solvents play a strategic role. To qualify as green media, the solvents must meet various criteria, such as being easily available, non-toxic, biodegradable, recyclable, economical, and non-flammable, among others [13]. Deep eutectic solvents (DESs) are defined as mixtures consisting of at least two components, one of which functions as the hydrogen bond acceptor (HBA) and the other plays the role of the hydrogen bond donor (HBD). The components of the mixture are capable of interacting with each other via hydrogen bonding to form a new eutectic phase characterized by a lower boiling point than that of each of the components separately [13,14]. Moreover, the system created is in a liquid form under ambient conditions. Typically, a halide salt and a hydrogen bond donor (HBD) are used for DES preparation, capable of forming a complex with the halide salt anion. The number of possible combinations of HBA and HBD capable of forming eutectic mixtures is extremely high, and many have been reported to date [15,16,17,18]. As for the significant advantages of DESs, they are easy to manufacture by simply mixing the ingredients in different molar proportions under heating and stirring. Usually, the components used to create DESs are of low cost, thus enabling them to be produced on a large scale [13,19,20,21,22]. Extraction using DESs could be viewed as a low-energy method, since extraction is usually performed at room temperature. There is no need to recover DES, because the whole extract could be incorporated in the food matrix. Moreover, DESs are considered eco-friendly, being classified as biodegradable materials [23].

Edible films are thin layers of materials that have the ability to form films (such as proteins, fats, carbohydrates, or their combinations), and are certainly edible and safe for consumption. They are placed on the surface of food products, and play a role in primary packaging [24]. Such films are mainly used to prevent the loss of moisture in foods, provide a semi-permeable barrier to the movement of solutes, oxygen, and CO_2_, and extend the shelf life by inhibiting chemical, biological, and microbial alterations in the product. They can also improve the organoleptic characteristics of foods by acting as carriers of active substances, such as dyes, flavor enhancers, antioxidants, spices, and flavorings [25,26].

Edible films and coatings are applicable to various types of food products, such as fresh agricultural produce, meat, fish, dairy, cereals, and lightly processed foods, such as peeled fruits and vegetables. Moreover, the increased interest in edible packaging is due to the consumer’s demand for safe, healthy, and sustainable food products that are naturally preserved, as well as the awareness of the negative environmental impacts of non-biodegradable petroleum-based packaging materials. Therefore, it is understood that research on edible packaging has environmental advantages over the conventional plastics, and is promising alternative in terms of food preservation [25,27]. In order to enhance the functional characteristics of manufactured biopolymer-based membranes, different agents are often used to improve their properties (mechanical behavior and barrier characteristics) and make them more attractive to consumers. The aforementioned agents can be additives that improve or modify the basic functionality of the material (e.g., plasticizers, amplifiers, and emulsifiers) and components that improve the quality, stability, and safety of the packaged food items [25,28,29].

Consequently, the aim of this study was to investigate the physicochemical properties of chitosan-based edible films containing pomegranate peel extracts as plasticizers. Novel deep eutectic solvents (DESs) were used as green extraction solvents of the pomegranate peels. Choline chloride and glycerol were employed as the hydrogen bond acceptor and hydrogen bond donor constituents of the binary solvent system, respectively.

## 2. Materials and Methods

### 2.1. Chemicals and Reagents

Chitosan was obtained from Seikagaku Corporation (Tokyo, Japan). Inorganic salts (reagent grade) used for adjusting the relative humidity in chambers employed to determine the water sorption behavior of the chitosan-based films were from Merck KGaA (Darmstadt, Germany). Folin–Ciocalteu was from Fluka (Steinheim, Germany). Gallic acid, glycerol (>99%), and choline cloride were from Sigma Chemical Co (St. Louis, MO, USA). Quercetin (quercetin dehydrate, 97%) was from Alfa Aesar GmbH & Co KG, whereas acetic acid (glacial 99–100% a.r.) was from Chem-Lab NV. All aqueous solutions were made using distilled water of HPLC quality.

### 2.2. Plant Material

Pomegranate peels (Wonderful variety) were obtained as a waste byproduct from a local fruit juice industry (Rodi Hellas, Greece). The peels were dried for 48 h at 40 °C and were subsequently ground in a mill (Janke and Kunkel, IKA Labortechnik, Germany). The initial moisture content was 75%, and the moisture content after drying was 8%. The distribution of peel particle size was determined by a sieve system (Retax, Labor Siebmaschine, Germany). The mean particle diameter was about 0.1 mm. The dried and ground material was stored in airtight glass containers prior to its use.

### 2.3. Extraction Procedure

Various amounts (from 0.65 to 2.35 g) of dried and ground pomegranate peel were mixed with 20 mL of DES (containing various water contents, from 5–55% *w/v*) and placed in a glass bottle with a stopper. The DES-ground peel mixture was then mixed in a water bath under stirring at 600 rpm for various times (from 36 to 103 min) at a constant temperature of 50 °C. After extraction, the samples were filtered and stored under freezing conditions until further analysis.

### 2.4. Total Polyphenol Yield

The concentration of total polyphenols in the extracts (CTP) was determined according to a well-established protocol [30], using the Folin–Ciocalteu method. The yield in total polyphenols (YTP) was expressed as mg of gallic acid equivalents (GAE) per g of dry pomegranate peel weight.

### 2.5. Total Flavonoid Yield

The protocol used for the determination of total flavonoid yield (YTF) was described by Surana et al. [31], with some modifications. The yields of total flavonoids were expressed as mg of quercetin equivalents (QEs) per g of dry pomegranate peel weight.

### 2.6. Experimental Design

A six-point central composite design has been adopted to test the impact of the independent variables on total polyphenol (YTP) and total flavonoid (YTF) content. More specifically, the three independent variables studied were the concentration of water in the DES mixture (Cw, % *v*/*v*), ranging between 5 and 55%, the liquid to solid ratio (R_L/S_, mL/g), ranging from 13 to 47, and the duration of extraction (t, min), ranging from 63 to 103 min. Each variable was coded at five levels: −1.682, −1, 0, 1, and 1.682 (Table 1). The three independent variables were coded according to the following equation:
(1)$$\;x_{i}=\frac{X_{i}-X_{0}}{\Delta {Χ}_{\iota }}\;\;\;\;\;\;\;x_{i}=1,\;2,\;3\;\ldots$$
where *x_i_* and *X_i_* are the dimensionless and real values of the independent variable *i*, respectively, *X*_0_ is the real value of the independent variable *i* at the central point, and *ΔX_i_* is the step of changing *X_i_*. The total polyphenol extraction yield (YTP) and total flavonoid yield (YTF) were selected as the dependent variables or responses, as they are known to be significantly affected by the extraction process [32,33]. The data obtained were subjected to regression analysis using the least squares methodology in order to derive the equations that express the response values as a function of the independent variables (mathematical models). Analysis of variance (ANOVA) was used to assess the statistical significance of the model. Insignificant dependent terms (*p* > 0.05) were omitted from the regression models obtained through a backward elimination process. Contour plots were finally obtained using the fitted model.

### 2.7. Statistical Analysis

All assays were performed in triplicate. The values given are the averages of the three measurements. The experimental design and surface methodology were performed with JMP™ 13 Pro (Cary, NC, USA).

### 2.8. Preparation of the Plasticized or Non-Plasticized Chitosan Films

The DES, consisting of choline chloride, glycerol (ChCl:Gly, 1:11), and the pomegranate peel extracts, were used as a plasticizer for the chitosan films; the peel extract contained 1235 mg of total phenolics/100 mL DES. The composition of each intervention is shown in the Table 2.

The chitosan film forming solution was prepared as follows. Initially, 2 g of chitosan were mixed with 100 mL of distilled water, and then 100 mL of aqueous acetic acid dilute solution (2% *v*/*v*) was added to the chitosan dispersion according to the protocol of [34] using a high-speed magnetic stirrer. Stirring was performed under ambient conditions until complete solubilization of the chitosan, followed by vacuum filtration to remove any insoluble chitosan residues. The filtrate was stored in the refrigerator for degassing until the day of use. When specimen composition required the use of a plasticizer, DES or DES extract was incorporated in the chitosan solution, followed by stirring and cooling until casting into plastic plates (multiple casting was carried out to obtain films with sufficient thickness). Chitosan films were then made by evaporation of the aqueous portion of the solution, with or without a added plasticizer. Drying was performed in an oven set at 33 °C and lasted for a few days. Depending on the composition of each formulation, the membranes were between 0.1–0.5 mm thick.

### 2.9. Film Characterization

#### 2.9.1. Sorption Behavior-Moisture Content

Portions of 3 × 3 × 0.02 cm^3^ and about 200 mg in weight were cut and dried at 100 °C for about 4 h after being placed in pre-weighed containers, and kept in chambers with different relative humidity (RH) for one week. The chambers contained saturated salt solutions to expose the samples to nine different relative humidity environments. At least three specimens from each film of a different composition were used to equilibrate in moisture with the nine different relative humidity environments. From the equilibrium moisture- water activity data, the isothermal adsorption curves (25 °C) were constructed. These data were adapted to the Brunauer-Emmett-Teller (BET) [35] and the Guggenheim-Anderson-DeBoer (GAB) [36] isotherm models.

#### 2.9.2. Large Deformation Mechanical Test (Tensile Strength)

Film specimens with dimensions 4 (length) × 0.8 (width) × 0.02 (thickness) cm were cut as a dog bone-type specimen with a width at the break point of the membrane of 0.6 cm, after equilibration at various relative humidity environments (chambers with saturated salt solutions at 25 °C), and subjected to large deformation tensile testing. The test was performed on a TA.XTplus texture analyzer (Stable Micro Systems, Godalming, Surrey, UK) at ambient temperature using two (A/TG) tensile grips where the sample was tied.

The texture analyzer software was employed to treat the force-deformation curve, calculating the maximum tensile strength, *σ_max_*, and the maximum elongation, E% (percentage elongation). These parameters are calculated by the following equations [37]:(2)σmax=Fmaxb×d
where *F_max_* is the maximum force that occurs when the sample is fractured, while *b* and *d* are the width and thickness of the specimen at the point of fracture, respectively, and:(3)$$\%E=\frac{\Delta L}{L_{0}}\times 100$$
where Δ*L* is the increase of the sample length at break and *L*_0_ is the original sample length, i.e., the distance between the analyzer clamps.

From the measurements of these parameters, the diagrams of *σ_max_* and %*E* vs. the moisture content of the pure polymeric and the spiked samples with DES and pomegranate peel DES extract were constructed. In this procedure, the reported *σ_max_* values represent the average of the measurements for at least eight samples in which the moisture content was equilibrated at each specified level; the moisture content of the samples used for the mechanical tests of large deformation was determined by drying at 100 °C to constant weight.

#### 2.9.3. Color Parameters

The color of the films was determined using a Konica Minolta colorimeter CR-400/410 (Kyoto, Japan). A white plate was used as the blank (Y = 97.59, x = 0.02, y = 1.79) to determine the CIE system parameters L, a, and b, where L is the parameter indicating lightness and *a* and *b* are parameters indicating the color (*a* red–green and *b* yellow–blue) of each film. The total difference in color is calculated using the equation:(4)$$\Delta Ε=\sqrt{{\left(L-L^{*}\right)}^{2}+{\left(a-a^{*}\right)}^{2}+{\left(b-b^{*}\right)}^{2}}$$
where *L**, *a**, and *b** represent the respective values of the white plate surface on which the calibration was performed.

All measurements were made seven times at different points in each film, and the values for each film of different composition constituted the averages of the mean values of three different membranes at each film formulation.

Statistical processing for color differences between the different film compositions (Ch, 25DES, 50DES, 25DES0, and 50DES0) was carried out with ANOVA (Tukey test) using SPSS (IBM SPSS, v.25.0. Armonk, NY, USA: IBM Corp.).

## 3. Results

### 3.1. Optimization of the Extraction Process

The extracts from dried and powdered pomegranate peels were stirred at 600 rpm at 50 °C. The procedure used for the extraction study aimed at examining the effect of the three selected variables, which were the concentration of water in the DES (C_W_, % *v/v*), the ratio of liquid to solid (R_L/S_, mL/g), and the duration of extraction (t, min), on the yield of the extraction. The values for the dependent variables (YTP and YTF) were analyzed experimentally by multiple regression, and, after the removal of the non-significant terms (*p* > 0.05), the theoretical models obtained are shown in Table 3.

The reliability of the model was assessed using the square of the correlation coefficient (R^2^), which was over 0.89 (*p* < 0.05). The results indicated that there was a statistically significant correspondence between the observed and predicted values, and that the models given in Table 3 can predict the optimal experimental conditions with high reliability. The variation in YTP and YTF depended on the simultaneous variation of the process variables, as illustrated by the respective contour diagrams (Figure 1 and Figure 2).

For the overall extraction yield of polyphenols (YTP), the important variables of the mathematical model were the ratio of liquid to solid, while the water concentration in the DES had a negligible effect (*p* > 0.05). As for the total flavonoids (YTF), all of the variables studied seemed to have a significant contribution. The contour diagrams of Figure 1 and Figure 2 show the effect of the independent variables on the extraction yield of total polyphenols (YTP) and total flavonoids (YTF), respectively.

More specifically, when the ratio of liquid to solid (R_L/S_) was about 20 mL (solvent) per gram (plant material), in 90 min and at a 30% *v/v* water concentration in the solvent, a maximum polyphenol yield appeared. Simply, the yield in total phenolics decreased as the R_L/s_ increased, and was enhanced when the extraction period was 90 min and when the solvent contained 30% *v/v* of water.

Based on the contour diagrams of Figure 2, with 47 mL of DES per gram (liquid to solid ratio) and about a 70 min duration of extraction, as well as at approximately 30% *v/v* water content in the solvent, a maximum yield of total flavonoids occurred. The pH of the DES was 6.87. In that pH, phenolics that can be ionized (e.g., phenolic acids) are in their ionized form. Possibly, some ionic interactions also contribute to their solubilization during the extraction step.

In previous studies, a total phenolic content of pomegranate peels from three varieties ranging from 264 to 430 mg GAE g^−1^ dw and a total flavonoid content from 32 to 60 mg of rutin equivalent g^−1^ dw using 80% aqueous ethanol as the extraction solvent were reported [38]. In another study, ethanol extraction yielded a total phenolic content of 139.40 mg GAE g^−1^ dw [39], while similar results (196.7 mg GAE g^−1^ dw) were obtained by an extraction method using continuous high voltage electrical discharge [40]. Extraction with 80:20 methanol:water yielded total phenolics of 101.85 mg GAE g^−1^ dw, whereas the flavonoids in pomegranate extracts, as quantified by HPLC-DAD-ESI-MS and being present in insoluble, esterified, and free forms, were ~14.14 μg g^−1^ dw [41]. These authors noted that most of the identified flavonoids were also present in pomegranate juice and seeds; in other studies [11] a total phenolic yield in pomegranate peel extracts of 423.5 ± 31.8 mg GAE g^−1^ dw when extracted with methanol and 226.9 ± 13.5 mg GAE g^−1^ dw with water as the solvent have been reported. Using enzymes combined with supercritical fluids the yield of the total phenolics appeared to reach 301.53 ± 7.86 mg GAE g^−1^ dw [42]. Furthermore, in a study that tested the effectiveness of extracting phenolic substances from whole pomegranate fruit using an aqueous solution of β-cyclodextrin, the total polyphenolic content was 150.6 mg GAE g^−1^ dw, and the antioxidant activity of the extract was 46.24 ± 7.33 mg TRE/mL.. Overall, the results of these studies indicate that, although pomegranate peels are a common byproduct in the production of pomegranate juice, they can be used as raw materials for the preparation of extracts enriched with polyphenols [4].

In recent years, there has been a lot of research on the extraction of bioactive substances from various plant materials, mainly from byproducts of the food industry, with green solvents, such as various combinations of deep eutectic solvents (DESs). The maximum total phenolic content observed in the present work under optimal conditions was 272.98 mg GAE g^−1^ dw, and was derived from extraction of pomegranate peel powder with the DESs consisting of choline chloride (ChCl) and glycerol (Gly) at a ratio of 1:11. A very recent study by Rajh et al. [43] elaborated the potential use of infrared as an another efficient method for extracting polyphenols from pomegranate peels, yielding 152 mg GAE g^−1^ dw; this result was obtained with DES (malic acid:glycerol:glucose at 1:1:1) compared to extraction with water or 50% aqueous ethanol as solvent using sonication In a previous study by Mouratolgou et al. [44], different extraction solvents of phenolics from various plant byproducts of the food industry (red grape pulp, olive leaves, wheat bran, coffee, lemon peels and onion byproducts) were compared using 60% aqueous ethanol, water, and various combinations of DESs. The results showed that the plant material for which aqueous ethanol was the most efficient extraction solvent was olive leaves, while, for all other materials, the glycerol/choline solvent (3:1) proved to be the most effective. The yield of total polyphenols (YTP) achieved for lemon peels with the latter solvent (53.76 mg GAE g^−1^ dw) was much higher than that reported in the literature for ultrasound-assisted extraction (18.12 mg GAE g^−1^ dw). For the rest of the plant materials, the yields observed were very close to values found in the literature. Compared to the above findings, the data regarding total flavonoid yield (YTF) from lemon and onion peels, as well as wheat bran showed that none of the DESs examined were efficient to enhance the yield [44,45]. Regarding the extraction of oil mill waste, an 18% higher value was noted when the DES glycerol/glycine (YTP 106.25 mg GAE g^−1^ dw) was employed as a solvent compared to a 60% aqueous ethanol solution [46]. A similar study was carried out involving aromatic medicinal herbs, such as fennel, marjoram, sage, and dittany (*Origanum dictamnus* L.); lactic acid-based NADES (natural deep eutectic solvents) were used in this work [47]. The results showed that the solvent made with a combination of lactic acid/glycine/water (3:1:3) resulted in a high extraction yield for polyphenolic substances, and the combinations of lactic acid/sodium acetate (3:1) and lactic acid/ammonium acetate (3:1) were just as effective. The results obtained for extraction of total flavonoids with the NADES lactic acid/glycine/water system indicated a maximum efficiency only for dittany, while it was the least efficient for fennel. Furthermore, this combination of substances in the NADES much higher yield of total phenolics for marjoram (137.36 mg GAE g^−1^ dw) was feasible by ultrasonic extraction, compared to the rather low levels reported in the literature using other conventional extraction schemes (26.20 mg GAE g^−1^ dw) [47].

The use of new green solvents seems to serve the current interest and demand for green processes aimed at recovery of bioactive compounds from natural products and agricultural industrial waste residues, contributing to the emerging targets of a cyclic economy framework [48,49,50]. Since the positive characteristics of DES and NADES are related to the solvent composition, an optimization scheme is necessary because of the variety of cations, anions, HBA (hydrogen bond acceptor), and HBD (hydrogen bond donor) used to formulate such solvents. In addition, the regeneration and recycling of these solvents in order to reduce operating costs and improve the overall environmental impact are processes that require further research, as well as to explore combination of processes in optimizing extraction efficiency. It is important to note that sugar-based DESs appear to keep phenolic compounds stable, suggesting their potential application as solvents in the cosmetics and food industries [51]. Moreover, alternative extraction methods with DES and NADES can be performed, involving supercritical fluid extraction (SFE) processing or enzyme-assisted extraction (EAE) treatments [48]. 

### 3.2. Membrane Water Adsorption

Adsorption isotherms provide information on how different materials behave when exposed to various relative humidity environments. Hydrocolloids such as chitosan can adsorb large amounts of water in environments of high relative humidity due to their hydrophilic nature. Hydrophilic plasticizers also behave similarly, and enhance water absorption of a polymeric hydrocolloid matrix. As shown in the diagrams of Figure 3, the isotherms of the films made of chitosan and its mixtures with the DES solutions have a sigmoidal shape.

At low water activities, relatively small amounts of water were adsorbed (up to the water activity of 0.5–0.6), followed by a sharp increase in the water content of the membranes at higher relative humidity environments. The sorption isotherms of Figure 3 show that pure chitosan films adsorbed less water than all of the other composite films (chitosan combined with DES or the DES-extracted material). In a 94% RH environment, the largest amount of water seemed to be adsorbed from the composite chitosan membranes with 50% DES (51.21 g of water/100 g dry film material) or 50% DES pomegranate peel extract, with a small difference between them. Membranes made with the DES or the DES pomegranate peel extract at a level of 25% were ranked between them, which is expected, as glycerol is an effective plasticizer of hydrophilic nature, compatible with hydrophilic polymeric matrices, like chitosan. Therefore, the higher the DES or DES-extract contents in the membrane preparation solution, the more water the composite membranes are expected to adsorb in humid environments [52]. In addition, comparing the membranes with the same level of plasticizing components, it seemed that the addition of pomegranate peel extract reduced the moisture adsorption capacity of the membranes, especially in environments with intermediate and high relative humidity values (aw > 0.4), most likely due to the hydrophobic nature of the extracted phenolics from the pomegranate peel.

Additional information on the water adsorption phenomena can be obtained by adapting the experimental isotherm data (Figure 3) to the BET and GAB sorption models. The importance of these models is great, as they predict the adsorption of water in most food systems [53]. The calculated monolayer moisture (mm) values and other parameters using the BET (K) and GAB (K‘, C) equations are listed in Table 4. In general, it appears that the experimental data fit well into the two mathematical models, as indicated by the high correlation coefficient values (R^2^ > 0.88). The range of estimated monolayer moisture (mm) valued for the samples tested varied between 5.3–9.1 and 6.6–12.8 g of water/100 g of dry material, using the BET and GAB sorption models, respectively. The monolayer moisture values calculated by both models followed the order Ch < 25DES < 25DES0 < 50DES < 50DES0 for the reasons mentioned above.

The Guggenheim constant C represents the total heat of sorption for the first layer of water molecules (monolayer) adsorbed, merely reflecting the bond strengths between water molecules and the hydrophilic sites in the polymer film matrix; i.e., the higher the C value, the stronger the interaction between the water molecules and the sorption sites of the solid material. In the present work, pure chitosan films exhibited the maximum value of C, followed by the plasticized film with 25% DES pomegranate peel extract, indicating that the water adsorption from these films is characterized by a monolayer of water molecules strongly attached to the material. For the remaining membranes that showed lower C values, it can be considered that the adsorption of water molecules as a monolayer was not as strongly favored as in the case of Ch or 25DES films [54].

### 3.3. Large Deformation Mechanical Test (Tensile Strength)

With the tensile tests, the force-deformation curves were recorded, from which the maximum tensile strength, *σ_max_*, and the maximum percentage elongation, Ε%, at break can be obtained. The effect of moisture on chitosan films, as well as on the composite chitosan films with added DES or DES pomegranate peel extract, is shown in Figure 4. The maximum tensile stress and percentage elongation are presented in Figure 4a and Figure 4b, respectively.

At low water contents, some of the samples were in the glassy state at ambient temperature at which the test was performed (e.g., 100% chitosan and the 25% DES fortified chitosan films). As a result, relatively high values for *σ_max_* were registered, along with relatively small extensibilities (maximum percentage elongation %*E*) at low water contents of the films. For these samples, at intermediate moisture contents, a large decrease in the *σ_max_* and a significant increase in the E% values were noted, which reflects the gradual transition of the material failure mode from hard and brittle to ductile. For chitosan films plasticized with water or other compatible low molecular weight plasticizers the stress-deformation curves indicated that the materials were in the elastoplastic state at room temperature. This phenomenon indicates the modification of the state of the film materials from glassy to plastic, and occurs at a moisture content that is influenced on the level of polyol (sorbitol) used in the composite film [55]. When the level of co-plasticizer increased in the chitosan membranes from 25 to 50% (in dry matter) in the mixtures, relatively small values in *σ_max_* and high %*E* were recorded at all moisture contents.

In Figure 4b, the curves describing the relationships between moisture and maximum percentage of elongation (%*E*) for the chitosan films are presented. It is interesting to note the distinct differences in %*E* at low water contents between the films plasticized with either 50% DES or 50% DES extract (more pronounced elongation) vs. pure chitosan or chitosan plasticized with 25% DES/DES extract. Moreover, with the 25% DES plasticizer level, the composite chitosan films were more easily softened by the absorbed water when the DES was incorporated alone (25DES0) compared with its counterpart containing the phenolic extract in the DES solvent (25DES); apparently, films with the latter composition began to plasticize at a much higher water content (Figure 4b), most likely due to the hydrophobic nature of the included phenolics in the composite film matrix.

### 3.4. Color Parameters

The chitosan and composite chitosan films with 25 and 50% *w/w* (dry matter) DES and DES pomegranate peel extract, acting as plasticizers, had the colors shown in Figure 5.

The membranes were yellow or pale yellow in color, and clearly, those containing the pomegranate peel extract had the most intense yellow tinge. All membranes were translucent as well as relatively homogeneous and flexible, with no visible pores, insoluble components, or cracks. Detaching them from the plastic plate was easy for all film formulations. The effect of DES and DES pomegranate peel extract on the color of chitosan membranes, as reflected in the parameters of L (brightness), a (red), and b (yellow), but also for the total color difference (ΔE), is presented in Table 5.

As expected, membranes with 100% chitosan and membranes with DES (consisting only of choline and glycerol) showed similar ΔE values, which were much lower than those of membranes where as a plasticizer a DES extract of the pomegranate peel was employed. Differences in the ΔE values were also seen between membranes fortified with 25 and 50 % DES containing the pomegranate extracts; i.e., the highest extract included in the film forming solution the greater the ΔE value. 

The addition of DES extract to the membrane preparation solution also increased the parameters a, b, and ΔE of the composite chitosan films, and it decreased the parameter L. This phenomenon has been previously noted by other authors [56] on chitosan membranes in which DES and curcumin were incorporated. Similar results have been also reported in other studies with chitosan membranes in which polyphenols from propolis, tea, or cinnamon essential oil have been included [57,58,59]. Pigments such as curcumin are responsible for the yellow color of polysaccharide edible films [60].

## 4. Conclusions

The present study showed that the combination of glycerol with choline chloride in a molar ratio of 11:1 can be an effective deep eutectic solvent (DES) for the extraction of phenolic substances from byproducts of pomegranate processing and, more specifically, the peel residue.

The results of the optimization of extraction showed that a specific DES combination used as an extraction solvent can compete with or even exceed (272.98 mg GAE g^−1^ dw) the values for phenolic contents reported in the literature regarding extracts from pomegranate peel or other agricultural waste materials, implying an effective extraction scheme for the preparation of phenolic bioactive preparations with potential applications in the food, medicine, and cosmetics industries. Incorporation of the DES choline chloride/glycerol alone and the DES of pomegranate peel extract into chitosan-based composite films modified the water sorption behavior of the mixed polysaccharide matrix. More specifically, membranes with more co-plasticizer (DES components) adsorbed more water in environments of high relative humidity. Membranes with the same level of DES used as plasticizer, the water adsorption was reduced when pomegranate peel extract was also included, possibly due to the hydrophobic nature of the phenolic substances. As for the mechanical properties of the prepared chitosan membranes, those made with either 100% chitosan or with the inclusion of 25% DES or 25% DES0 at low relative humidity environments (25 °C), were in a glassy state, exhibiting high values of *σ_max_* and low %*E* at break. In environments of intermediate relative humidity, these membranes switch to an elastoplastic state, a fact that is demonstrated by the strong reversal of the above values in the mechanical properties.

Finally, the color of the membranes studied was yellow or pale yellow, depending on the percentage and type of plasticizer that was incorporated each time in the chitosan film forming solution. Clearly, the pomegranate peel extract was the one that gave the strongest shade to the membranes, showing significant differences in the values of ΔE compared to the respective values obtained for the composite chitosan membranes without the inclusion of pomegranate extract.

## Figures and Tables

**Figure 1 foods-10-01262-f001:**
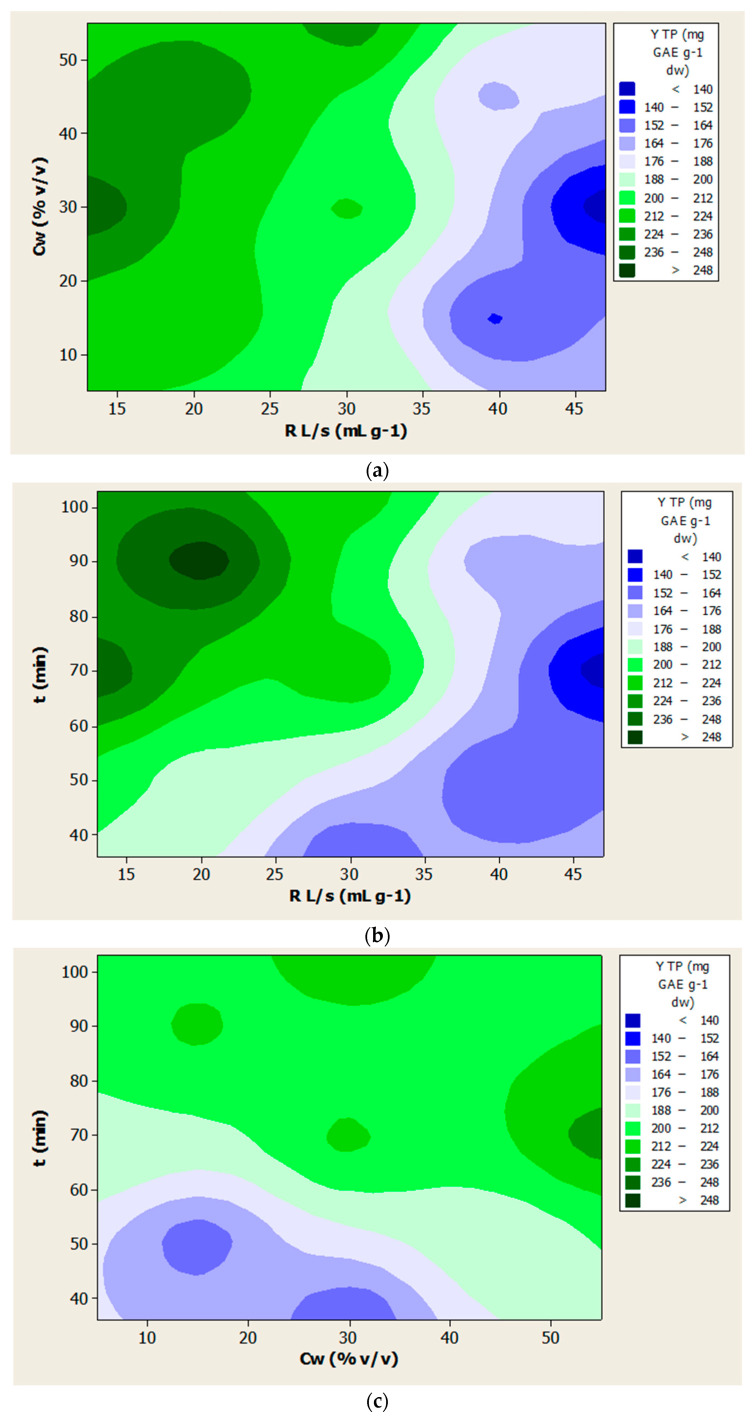
Contour diagrams (**a**–**c**) for the extraction yield of total polyphenols (YTP).

**Figure 2 foods-10-01262-f002:**
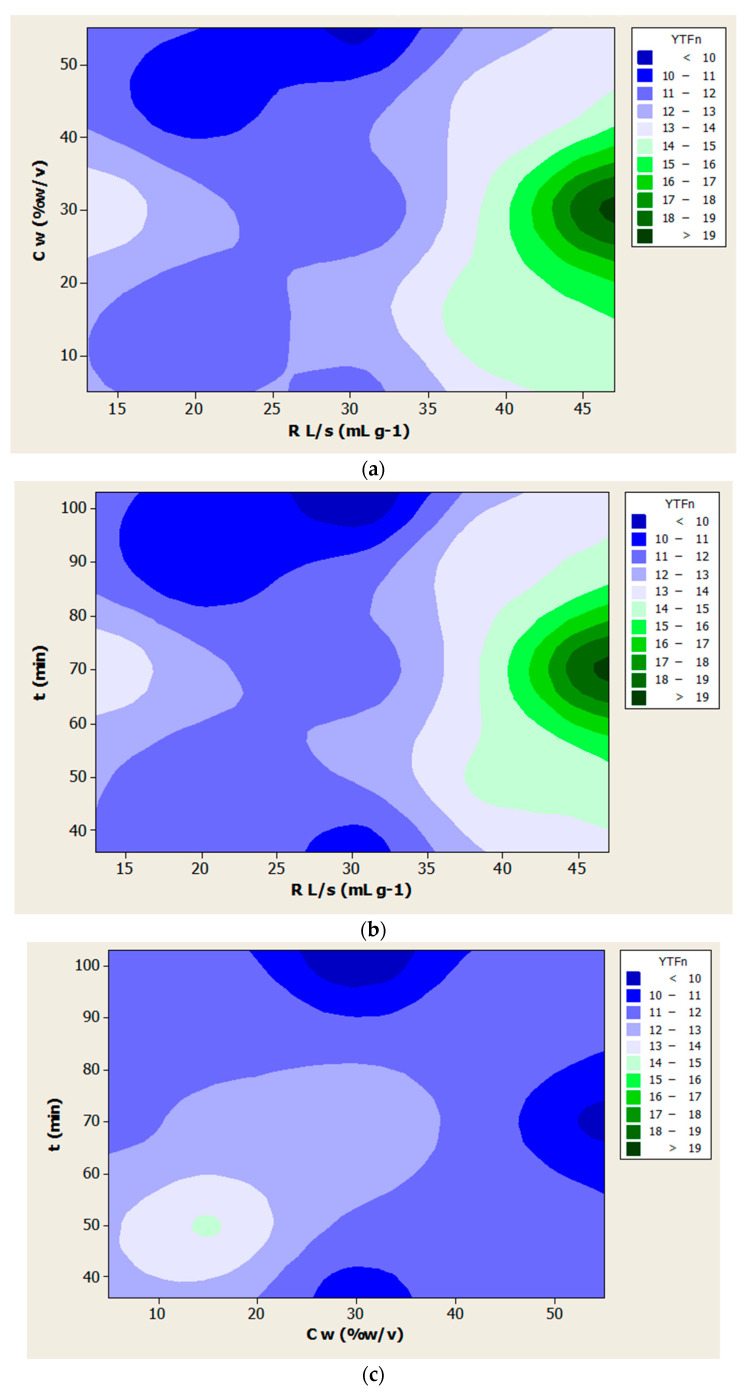
Contour diagrams (**a**–**c**) for the extraction yield of total flavonoids (YTF).

**Figure 3 foods-10-01262-f003:**
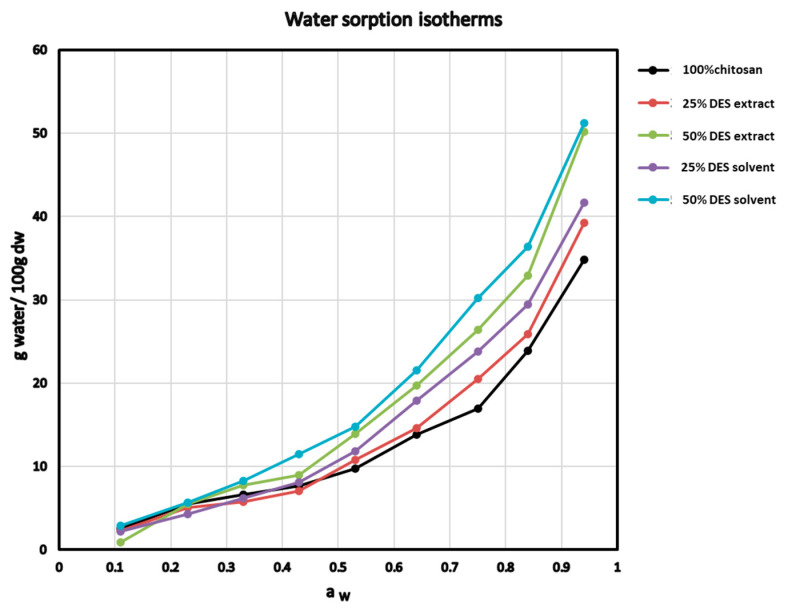
Water adsorption isotherms of chitosan composite films at 25 °C.

**Figure 4 foods-10-01262-f004:**
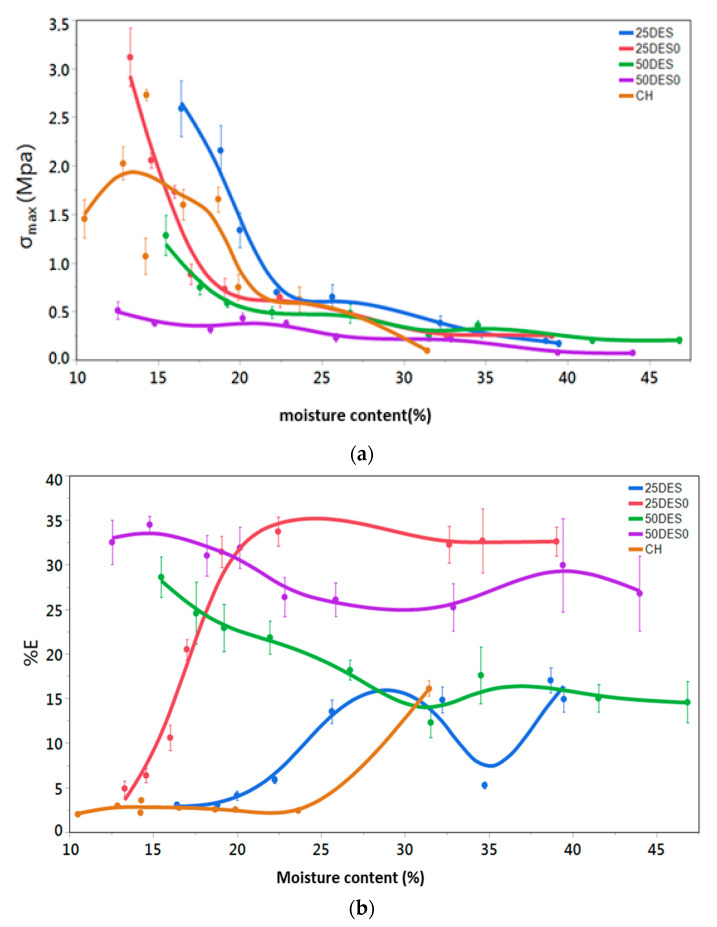
Τhe effect of moisture content of the membranes (**a**) on the maximum tensile stress (σmax) at the break and (**b**) on the percentage elongation (%*E*) at the break.

**Figure 5 foods-10-01262-f005:**
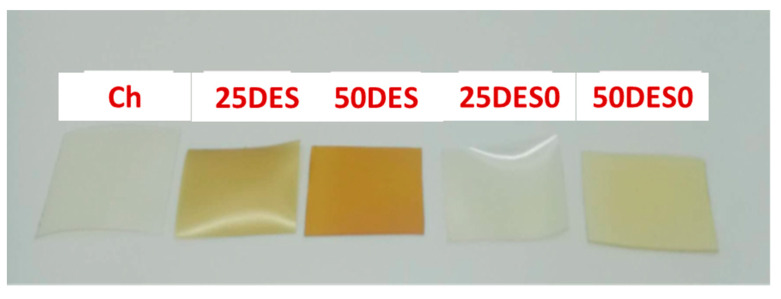
The color of the chitosan membranes with and without DES plasticizer and pomegranate peel extract in DES.

**Table 1 foods-10-01262-t001:** Experimental values and coded levels of the independent variables used for the central composite design.

Independent Variables	Coded Units	Levels of Coded Variables				
		−1.682	−1	0	1	1.682
C_W_ (%. *v/v*)	X1	5	15	30	45	55
R_L/S_ (mL g^−1^)	X2	13	20	30	40	47
t (min)	X3	36	50	70	90	103

**Table 2 foods-10-01262-t002:** Composition and symbolization of samples in the form of films.

Samples	Composition (% Dry Matter)		
	Chitosan	DES	DES Extract
Ch	100	-	-
25DES	75	-	25
50DES	50	-	50
25DES0	75	25	-
50DES0	50	50	-

**Table 3 foods-10-01262-t003:** Polynomial equations and statistical parameters calculated by applying a central composite design.

Dependent Variable	2nd Degree Polynomial Equations	R^2^	*p*
YTP (mg GAE g^−1^ dw)	221.50 − 30.78 × X2	0.93	0.0030
YTF (mg QE g^−1^ dw)	11.70 − 0.6 × X1 + 1.64 × X2 + 1.69 × (X2)^2^ − 0.65 × (X3)^2^	0.89	0.0119

**Table 4 foods-10-01262-t004:** Parameters of the BET and GAB adsorption models applied to moisture adsorption data (25 °C) of chitosan and chitosan membranes spiked with the DES (25DES0, 50DES0) and the DES pomegranate peel extract (25DES, 50DES).

Sample	BET (aw Range 0.11–0.64)			GAB (aw Range 0.11–0.94)			
	mm (g Water/100 g dw)	K	*r*^2^ (*n* = 6)	mm (g Water/100 g dw)	K’	C	*r*^2^ (*n* = 9)
Ch	5.3	7.9	0.98	6.6	0.87	5. 7	0.90
25DES	5.6	5.0	0.93	6.8	0.90	4.0	0.88
50DES	7.4	4.4	0.92	10.3	0.87	2.8	0.92
25DES0	6.4	3.5	0.94	9.6	0.86	2.3	0.95
50DES0	9.1	3.2	0.99	12.8	0.84	2.4	0.97

**Table 5 foods-10-01262-t005:** Color parameters of the various composite chitosan films (L, a, b and DE).

Samples	L	a	b	ΔΕ
Ch	91.76 ± 0.11 ^a^	−1.13 ± 0.07 ^b^	5.27 ± 0.30 ^c^	6.89 ± 0.25 ^c^
25DES	77.04 ± 1.09 ^b^	−0.79 ± 0.41 ^b^	38.00 ± 1.15 ^c^	41.65 ± 1.52 ^b^
50DES	76.43 ± 0.98 ^b^	1.52 ± 0.73 ^a^	44.76 ± 1.00 ^c^	47.93 ± 1.35 ^a^
25DES0	91.63 ± 0.24 ^a^	−1.04 ± 0.37 ^b^	4.70 ± 1.46 ^b^	6.82 ± 0.86 ^c^
50DES0	92.64 ± 0.64 ^a^	−1.11 ± 0.23 ^b^	4.98 ± 1.07 ^a^	6.03 ± 1.04 ^c^

Averages followed by a different letter differ significantly, *p* = 0.05 according to the Tukey test.

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
