# Peer review of "Physical Properties of Chitosan Films Containing Pomegranate Peel Extracts Obtained by Deep Eutectic Solvents"

_foods, 2021, doi:10.3390/foods10061262_

Round 1
Reviewer 1 Report
The manuscript studies physical properties of chitosan films containing pomegranate peel extracts obtained by deep eutectic solvents
In my opinion, the manuscript is well written, clear and understandable. The article has been done fairly, the information are satisfactory and indicate the correctness of the developed methods. The article is of great interest and can be useful for researchers.
Referee has only few remarks.
- Authors have to compare their DES solvents in terms in advantages and drawbacks, and detailed what are the green impacts : reduction of solvent, energy, wastes… This reference could help the authors
Review of Alternative Solvents for Green Extraction of Food and Natural Products: Panorama, Principles, Applications and Prospects. Molecules 2019, 24, 3007. https://doi.org/10.3390/molecules24163007
- It is known that depends on pH, some compounds will change their solubilization from hydrogen bonds to ionic bounds which are stronger and more solubilization occurred. Authors have to discuss the effect of pH and the addition of cosolvents.
Author Response
We thank Reviewer #1 for his/her comments.
Referee has only few remarks.
Comment 1:
- Authors have to compare their DES solvents in terms in advantages and drawbacks, and detailed what are the green impacts : reduction of solvent, energy, wastes… This reference could help the authors
Review of Alternative Solvents for Green Extraction of Food and Natural Products: Panorama, Principles, Applications and Prospects. Molecules 2019, 24, 3007. https://doi.org/10.3390/molecules24163007
Response:
Information regarding reduction of solvent, energy, wastes added in the introduction section. The suggested reference has been included in the article.
Comment 2:
- It is known that depends on pH, some compounds will change their solubilization from hydrogen bonds to ionic bounds which are stronger and more solubilization occurred. Authors have to discuss the effect of pH and the addition of cosolvents.
‘’The pH of the DES solvents was 6.87. In that pH phenolics that can be ionized e.g. phenolic acids are in their ionized form. Possibly, some ionic interactions also contribute to their solubilization during extraction’’ This information was added in the discussion section
Reviewer 2 Report
Application of new extraction technologies in the designing of new bio-based food packaging is a very interesting topic. My main concern is about the advantages of including the pomegranate extract (DES extracts) into the chitosan film rather than the DES alone (DES solvents). This should be better stressed in the paper.
The following minor revisions are suggested to the authors.
- Please, insert the initial and final moisture content of pomegranate peels in the section 2.2 (rows 100-101)
- Many studies have examined the pomegranate peels extract (PPE) using different extraction methods, such as enzymatic extraction, pressure-applied extraction, ultrasounds, and microwave-assisted extraction and some of them (especially using PUAE: pulsed ultrasound assisted extraction) have contributed to obtain PPEs with similar bioactive contents even in shorter time (5-10 minutes). I suggest citing and discussing them in the Introduction to justify the rationale of the present paper. Probably rows 264-285 could be moved in the Introduction too and then compared in the discussion section. As well as many other sentences in 3.1 citing literature data not strictly related with the discussion of the present paper's results.
- Since ellagitannins are considered the most active compounds in the pomegranate peels, their quantification in the optimised extracts would be quite useful. At least as punicalagin content, as marker compound, or as ellagic acid equivalents.
- The following sentence “The results of this study indicated that although pomegranate peels are a common by-product in the production of pomegranate juice, they can be used on their own for the preparation of extracts rich in polyphenols” (at rows 286-287) should be pluralised since all the cited studies state that concept.
- Is there a reason to choose the molar ratio of glycerol to choline chloride fixed at 11: 1 instead of studying it as an additional variable in DoE? Since the cited literature seem not report this ratio.
- The following three rows in the Conclusions “This section may be divided by subheadings. It should provide a concise and precise description of the experimental results, their interpretation, as well as the experimental conclusions that can be drawn. “should be deleted.
Author Response
Reviewer #2
Application of new extraction technologies in the designing of new bio-based food packaging is a very interesting topic. My main concern is about the advantages of including the pomegranate extract (DES extracts) into the chitosan film rather than the DES alone (DES solvents). This should be better stressed in the paper.The following minor revisions are suggested to the authors.
We thank Reviewer #2 for his/her comments.
Comment 1:
Please, insert the initial and final moisture content of pomegranate peels in the section 2.2 (rows 100-101)
Response:
The initial moisture content was 75% and the moisture content after drying was 8%. This information was added in the manuscript.
Comment 2:
Many studies have examined the pomegranate peels extract (PPE) using different extraction methods, such as enzymatic extraction, pressure-applied extraction, ultrasounds, and microwave-assisted extraction and some of them (especially using PUAE: pulsed ultrasound assisted extraction) have contributed to obtain PPEs with similar bioactive contents even in shorter time (5-10 minutes). I suggest citing and discussing them in the Introduction to justify the rationale of the present paper. Probably rows 264-285 could be moved in the Introduction too and then compared in the discussion section. As well as many other sentences in 3.1 citing literature data not strictly related with the discussion of the present paper's results.
Response:
Introduction section rewritten. Rows moved from the discussion section to the introduction section as proposed.
Comment 3:
Since ellagitannins are considered the most active compounds in the pomegranate peels, their quantification in the optimised extracts would be quite useful. At least as punicalagin content, as marker compound, or as ellagic acid equivalents.
Response:
We agree with the comment. The quantification of ellagitannins it was not the target of this work as total polyphenols was one of the variables measured with response surface methodology
Comment 4:
The following sentence “The results of this study indicated that although pomegranate peels are a common by-product in the production of pomegranate juice, they can be used on their own for the preparation of extracts rich in polyphenols” (at rows 286-287) should be pluralised since all the cited studies state that concept.
Response:
The paragraph corrected according to reviewer suggestion
Comment 4:
Is there a reason to choose the molar ratio of glycerol to choline chloride fixed at 11: 1 instead of studying it as an additional variable in DoE? Since the cited literature seem not report this ratio.
Response:
The ratio selected in order to obtain DES solvent with a proper viscosity that can be handled
Comment 5:
The following three rows in the Conclusions “This section may be divided by subheadings. It should provide a concise and precise description of the experimental results, their interpretation, as well as the experimental conclusions that can be drawn. “should be deleted.
Response:
Conclusion section was re-written. Inappropriate paragraph removed.